# Historical visit attendance as predictor of treatment interruption in South African HIV patients: Extension of a validated machine learning model

Rachel T. Esra[1,2]*, Jacques Carstens[3], Janne Estill[1], Ricky Stoch[4], Sue Le Roux[5], Tonderai Mabuto[5], Michael Eisenstein[5], Olivia Keiser[1], Mhari Maskew[6], Matthew P. Fox[6,7], Lucien De Voux[3], Kieran Sharpey-Schafer[3]

1 Institute of Global Health, University of Geneva, Geneva, Switzerland, 2 Imperial College of London, London, United Kingdom, 3 Palindrome Data, Cape Town, South Africa, 4 Studio Fundi Ltd, London, United Kingdom, 5 The Aurum Institute, Johannesburg, South Africa, 6 Health Economics and Epidemiology Research Office, Department of Internal Medicine, School of Clinical Medicine, Faculty of Health Sciences, University of the Witwatersrand, Johannesburg, South Africa, 7 Departments of Epidemiology and Global Health, Boston University School of Public Health, Boston, Massachusetts, United States of America

* r.esra20@imperial.ac.uk

**Data Availability Statement:** Access to primary data is subject to restrictions owing to privacy and ethics policies set by the South African

## Abstract

Retention of antiretroviral (ART) patients is a priority for achieving HIV epidemic control in South Africa. While machine-learning methods are being increasingly utilised to identify high risk populations for suboptimal HIV service utilisation, they are limited in terms of explaining relationships between predictors. To further understand these relationships, we implemented machine learning methods optimised for predictive power and traditional statistical methods. We used routinely collected electronic medical record (EMR) data to evaluate longitudinal predictors of lost-to-follow up (LTFU) and temporal interruptions in treatment (IIT) in the first two years of treatment for ART patients in the Gauteng and North West provinces of South Africa. Of the 191,162 ART patients and 1,833,248 visits analysed, 49% experienced at least one IIT and 85% of those returned for a subsequent clinical visit. Patients iteratively transition in and out of treatment indicating that ART retention in South Africa is likely underestimated. Historical visit attendance is shown to be predictive of IIT using machine learning, log binomial regression and survival analyses. Using a previously developed categorical boosting (CatBoost) algorithm, we demonstrate that historical visit attendance alone is able to predict almost half of next missed visits. With the addition of baseline demographic and clinical features, this model is able to predict up to 60% of next missed ART visits with a sensitivity of 61.9% (95% CI: 61.5–62.3%), specificity of 66.5% (95% CI: 66.4–66.7%), and positive predictive value of 19.7% (95% CI: 19.5–19.9%). While the full usage of this model is relevant for settings where infrastructure exists to extract EMR data and run computations in real-time, historical visits attendance alone can be used to identify those at risk of disengaging from HIV care in the absence of other behavioural or observable risk factors.

Government. Data extraction, data anonymization and data management were approved by the University of Witwatersrand (Human Research Ethics Committee, Reference: 210106). The contact point for this data sharing agreement is the University of Witwatersrand Ethics Administrators (HREC-Medical.ResearchOffice@wits.ac.za).

**Funding:** The Aurum Institute is funded by the PEPFAR programme under grant number GGH001981: Programmatic Implementation and Technical Assistance for HIV/AIDS & TB Programs in Priority Districts of South Africa. The contents are the responsibility of the authors and do not necessarily reflect the views of PEPFAR, USAID or the United States Government. Jacques Carstens received funding in the form of salary from the commercial company Palindrome Data. Palindrome Data is partially funded by Janssen Pharmaceutica (Pty) Ltd, part of the Janssen Pharmaceutical Companies of Johnson & Johnson. The contents are the responsibility of the authors and do not necessarily reflect the views of Janssen Pharmaceutica (Pty) Ltd. The funders had no role in study design, data collection and analysis, decision to publish, or preparation of the manuscript. The corresponding author had full access to all the dataset in the study and had final responsibility for the decision to submit for publication. The specific roles of these authors are articulated in the 'author contributions' section.

**Competing interests:** Jacques Carstens received funding in the form of salary from the commercial company Palindrome Data. Palindrome Data is partially funded by Janssen Pharmaceutica (Pty) Ltd, part of the Janssen Pharmaceutical Companies of Johnson & Johnson. There are no patents, products in development or marketed products associated with this research to declare. This does not alter our adherence to PLOS ONE policies on sharing data and materials.

## Introduction

While South Africa has the largest HIV treatment programme globally, it is currently estimated that a quarter of the 7.5 million people living with HIV (PLHIV) are not on antiretroviral treatment (ART) [1]. ART is lifelong and stopping treatment results in rapid viral rebound, putting patients at an individual risk for AIDS-defining illness and increasing the risk of viral transmission [2]. Retention on ART remains a challenge in South Africa where 11–28% of patients become lost to follow up (LTFU) within the first two years of treatment initiation [3, 4].

ART treatment interruption in South Africa is likely mediated by a complex mix of socio-behavioural factors including mobility, stigma and health facility access [5, 6]. Cohort studies indicate that the risk of LTFU varies over time [7, 8] and many patients iteratively transition in and out of treatment [9], making behavioural drivers of ART retention difficult to define longitudinally. Without socio-behavioural information linked to routine HIV management, many retention interventions are focused on broad demographic sub-populations with perceived elevated rates of LTFU, including men, those diagnosed with HIV at younger ages and those initiating treatment with lower CD4 counts [10, 11]. However, little evidence supports the effectiveness of this approach [12, 13].

Innovative approaches to understanding and addressing risk of disengagement from HIV care are needed. Traditional statistical methods such as regression and survival analysis are frequently used to enumerate factors that describe elevated risk of LTFU [7–11]. Though widely adopted due to their ease of computation and explainability, these methods are limited in terms of accurately modelling collinearity, interaction effects and non-linear relationships between predictors [14] and are thus unable to uncover the complex mechanisms driving risk of disengagement from care. In contrast to this, machine learning methods are able to account for non-linear patterns often present in routinely collected observational data, and are increasingly being used to identify high risk subgroups of populations with suboptimal HIV service utilisation in low- and middle-income contexts [15–18].

We have previously described a machine learning algorithm able to predict up to two thirds of missed ART clinic visits using only visit attendance and routinely collected clinical information [16, 17]. In this model, patterns of historical visits attendance ranked higher than baseline demographic and clinical characteristics when predicting next missed visits [17]. While this model is able to predict the risk of disengagement from care at the level of on an individual patient and visit, the approach is still limited in terms of ability to infer relationships between predictors and interpret both the individual and relative role of potential predictors of treatment interruptions [17].

Previously, we identified 13 predictors for ART treatment discontinuation relating to age, baseline clinical characteristics and patterns of visits attendance from routinely collected ART patient records [17]. Here, we aim to expand the explainability of these predictors as a means of providing more generalised descriptions of the population at risk of IIT and the underlying drivers of risk. We assess the relative contribution of historical visit attendance in predicting risk of treatment interruption, by defining mutually exclusive and collectively exhaustive visit attendance archetypes encompassing this information in a single categorical variable. We then evaluate the predictive ability of the archetypes alone and in combination with the previously identified demographic and clinical predictors using both machine learning and traditional statistical methods.

## Methods

### Ethics

This study utilises routinely collected patient record data from the TIER.net electronic medical register (EMR), consisting of patient-level data collected at public health facilities providing

HIV care and treatment to the public sector in South Africa [19]. Data extraction, data anonymization and data management were approved by the University of Witwatersrand (Human Research Ethics Committee, Reference: 210106). Data extraction and anonymisation was performed by collaborators from The Aurum Institute South Africa, a not-for-profit organisation funded by the President's Emergency Plan for Aids Relief (PEPFAR) to support the implementation and improvement of ART services at the health facilities included in this study. The use of de-identified routine programme data to identify areas for quality improvement efforts is standard practice in South Africa, and critical for achieving the country's goals to control the HIV epidemic.

## Data sources and study participants

Our cohort included patients receiving ART from facilities from the Gauteng and North West provinces of South Africa. We included patient records from 1 January 2017, after the date of the implementation of the treatment for all policy, whereby ART initiation in South Africa was implemented for HIV patients regardless of HIV disease progression [20]. We included all patients newly initiated onto ART from the study start date to 24 March 2022, aged 15 years and older at ART initiation with a minimum of 18 months observation time. Based on cohort data indicating that the risk of LTFU stabilises after two years on treatment [21], person time was censored at 2 years after ART initiation. From the 264,635 patients that matched our inclusion criteria, we excluded patients who had died (0.03%, N = 8,028) or had transferred out to other facilities (23%, N = 61,775). We additionally excluded patients with records flagged as poor-quality including patients confirmed as LTFU at visits prior to final visits on record (N = 805) and patients with HIV diagnosis recorded after ART start date (N = 1).

## Measures

**Operational definition of outcomes.** For the purpose of this study, we considered different operational definitions of treatment interruption on the individual patient level. We assessed longitudinal treatment attendance on a visit by visit basis, by classifying each visit in a patient's visit trajectory as an interruption in treatment (IIT) if the visit was attended more than 28 days after the scheduled visit date [16, 17, 22]. On a patient level, we investigated the relationship between the longitudinal pattern of visit attendance and a final outcome of patient retention, where patients were considered LTFU if they were 90 days or more late for a scheduled visit at the end of our observation period in accordance with the South African Department of Health guidelines [20].

**Visit attendance archetypes.** In our previous work, variables describing historical visit attendance including the ratio of visits attended late vs. visits attended on time and the number of historical IITs, were shown to be more important in predicting next missed visits than baseline demographic and clinical features [17]. Based on these results and input from the clinical and program teams at The Aurum Health Institute, we developed mutually exclusive and collectively exhaustive visit attendance archetypes that describe historical visit attendance in a single categorical variable (Fig 1). For each visit attended, we defined visits attended within 14 days of a scheduled appointment to be on time and visits attended between 14–28 days of a scheduled appointment to be late (Fig 1). Using these definitions of visits attended on time, visits attended late and IITs, we defined visit archetypes as illustrated and described in Fig 1 and Table 1.

**Data analysis.** As described previously, clearly describing and explaining relationships between predictors in machine learning algorithms is difficult due to non-linearity and the collinearity [17]. While our previous work ranked historical visit attendance highly in predicting

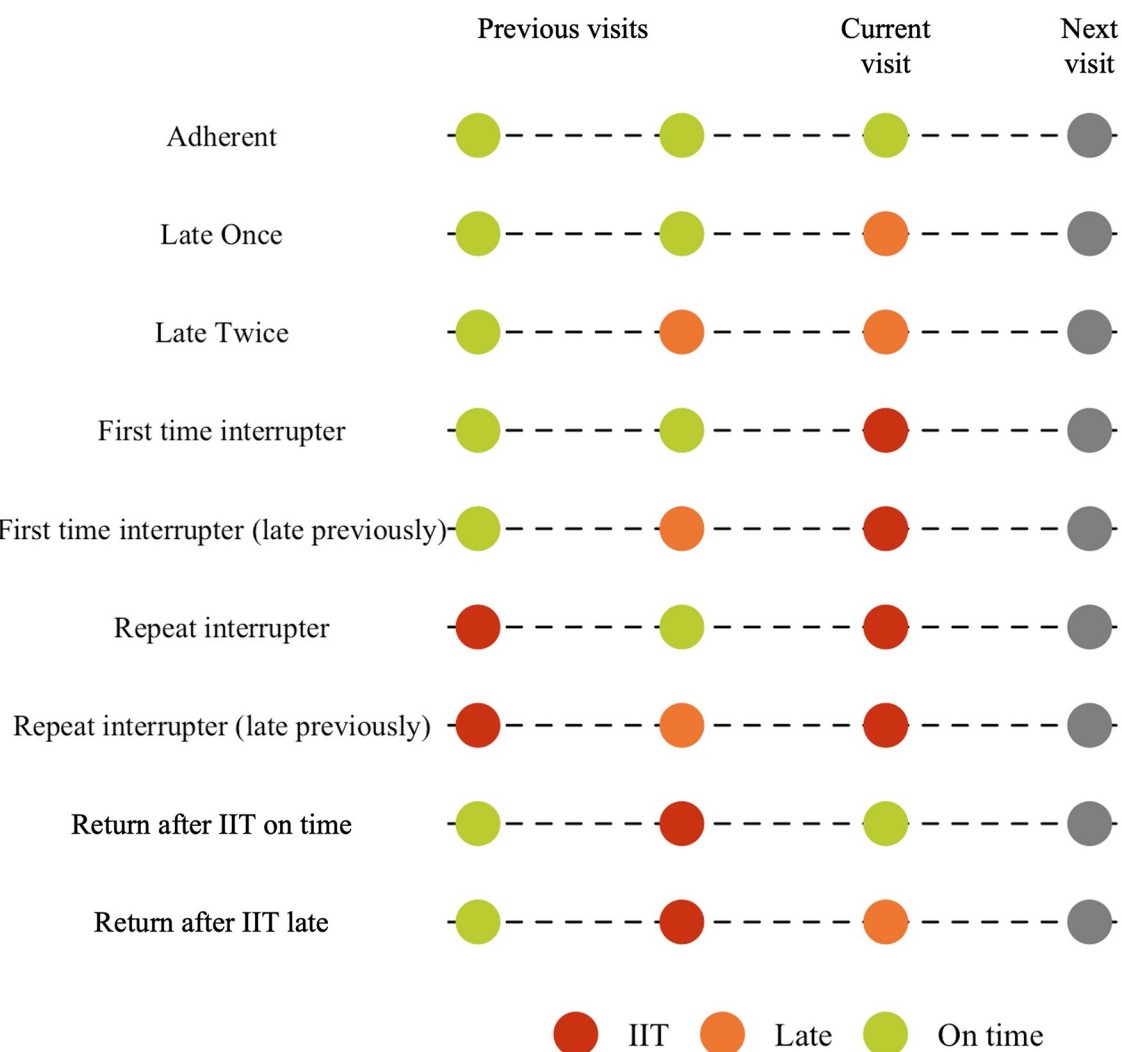

**Fig 1. Visit archetypes based on longitudinal patterns of ART visit attendance as described in Table 1.** Archetypes are mutually exclusive, completely exhaustive and defined by the historical pattern of visit attendance of interruptions in treatment (red), late attendance (orange) and visits attended on time (green). Analysis is focused on how these historical patterns are able to predict attendance at the next visit in the time series (grey).

next missed ART visits, using machine learning methods alone, we are unable to assess the relative contribution of historical visit attendance to other variables included. Here we evaluate the individual and combined predictive ability of historical visits attendance, baseline demographic and clinical characteristics comparing traditional statistical approaches with machine learning methods.

## Description of baseline and time varying risk factors

Descriptive statistics were used to characterise the demographic and clinical profile of patients at baseline and/or specific time points after ART initiation. We evaluated the demographic and clinical patient characteristics previously identified as predictive for IIT including sex, gender, age at ART initiation and baseline CD4 [17]. In order to adjust for changes in ART service delivery due to the Covid-19 pandemic [23], we included a binary variable describing the

**Table 1. Visits archetype definitions based on longitudinal patterns of ART visit attendance.**

| Visit archetype | | Definition |
|---|---|---|
| Adherent | On time | Current visit and previous visit attended on time OR Current visit attended on time and previous visit attended late |
| Late | Late Once | Current visit attended late and previous visit attended on time |
| | Late Twice | Current visit attended late and previous visit attended late |
| Interrupter | First time interrupter | First IIT (visit attended >28 days after scheduled visit date) after last visit attended on time |
| | First time interrupter late previously | First IIT (visit attended >28 days after scheduled visit date) after last visit attended late |
| | Repeat interrupter | Visit attended >28 days after scheduled visit date and patient has historical IIT and last visit |
| | Repeat interrupter late previously | Visit attended >28 days after scheduled visit date, patient has historical IIT and last visit late |
| | ITT twice | Visit attended >28 days after scheduled visit date and previous visit IIT |
| Returning defaulter | Visit after ITT on time | Visit on time where previous visit was an IIT |
| | Visit after ITT late | Late visit where previous visit was an IIT |

timing of ART initiation as preceding or during the national lockdown starting on 27 March 2020.

Defining baseline as the time of treatment initiation, we identified baseline risk factors using multivariable log binomial regression. We evaluated two separate outcomes, the risk of IIT and the risk of LTFU by the end of our observation period. For the latter, we included only patients who had two or more clinical visits. For all analyses, we report both sex aggregated and sex stratified estimates.

Second, we evaluated the impact of previous visit attendance on the time varying risk of IIT and LTFU using a non-parametric mixed effects Cox proportional hazards model. We included previous visit archetypes (Fig 1) and covariates identified as significant in our previous analysis in a model specified as:

Where the hazard of the occurrence the event at time for individual is the product of the baseline hazard, an exponentiated random effect for unobserved individual variance and linear function of predictors that may be time-invariant (,e.g. sex) or time-varying (, e.g. previous visits attendance). This semi-parametric extension of the cox proportional hazards model, violates the assumption of proportional hazards with inclusion of time varying covariates to account for within-subject correlation whereby the occurrence of an event may impact the occurrence of future events. We additionally include an individual level random effect, describing unmeasured heterogeneity in excess risk for clusters of individuals that cannot be explained by the observed covariates. In this recurrent event analysis, each IIT experienced was recorded as an event and patients who did not experience an IIT were censored at the end of the two year observation period. All statistical analysis was run in R version 4.2.1.

## Inclusion of visit archetypes in machine learning model

We have previously developed and validated a machine learning model predicting missed ART visits using baseline characteristics, historical visits attendance, clinical data and ART dispensing information constructed from the same South African EMR source [17]. Here we compare the performance of the model using the original set of 13 predictors, the original set of 13 predictors with previous visit archetypes, and previous visit archetypes alone. We apply the model to the same dataset with an extended study period. Model validation, feature

engineering and feature selection has been described previously [17]. Briefly, we randomly split 70% of visits into a training dataset (N = 1,833,248 visits) with the remaining 30% (N = 456,472 visits) reserved to act as an unseen test dataset. The training dataset was upsampled using the RandomOverSampler method from imblearn [24] to build a 50:50 balanced dataset. We implemented a gradient boosting model using the CatBoost algorithm [25]. The model was run for 1000 iterations using the model training parameters summarised previously [17].

Model performance was assessed using metrics to ascertain the ability to classify both the positive and negative outcomes. These included positive predictive value (PPV—proportion of predicted missed visits that were truly missed) and negative predictive value (NPV—proportion of predicted attended visits that were truly attended). We additionally evaluated the overall model performance, reporting the Area Under the model Precision Recall Curve (PR AUC, demonstrating model sensitivity and PPV at different classification thresholds), accuracy (total proportion of correctly identified visits) and F1 score (harmonic mean of overall model precision and recall). We constructed 95% confidence intervals using bootstrap resampling. We resampled the test dataset with replacement n = 1000 times, while the training set and model remained fixed. Feature importance was calculated using the Loss Function Change from CatBoost [25].

## Comparison of survival analysis to machine learning predictions

For each visit in the test dataset, the model above calculates a probability that an IIT will occur at the next visit. If the probability is higher than 0.50, the visit is assigned an outcome of predicted IIT. Model predictions are compared to occurrence of the outcome in the dataset and the model metrics are calculated accordingly. We compared the correlation between previous visit type and the predicted probabilities of IIT produced by the machine learning model, to the hazard ratios produced by survival analysis in the first part of this study.

## Results

### Cohort characteristics

Our cohort included 191,162 patients of which 63% were women and the median age of ART initiation was 34 years old (IQR: 28–41). Despite our study beginning after the implementation of treatment for all policies [20], only 55% of patients were initiated onto treatment on the same day of HIV diagnosis, 20% within a week of HIV diagnosis, 9% within two weeks diagnosis and 16% two or more weeks after HIV diagnosis.

### Baseline and time-varying risk factors for LTFU

Using the definition of a missed last appointment by 90 days or more, 38.8% (N = 73,978) of patients were defined as LTFU within two years of ART initiation. Of those that became LTFU, 24.5% (N = 18,568) did not return to treatment after initiation, 25.5% (N = 18,878) became LTFU within the first six months of treatment, 18.9% (N = 13,960) became LTFU between 6 months—1 year and the remainder 31.1% (N = 23,016) dropped out in the second year of treatment (Fig 1).

Overall, men were at a higher risk of LTFU after initiation (RR: 1.19 [95% CI: 1.15–1.23]) and within the first two years of treatment (RR: 1.07 [95% CI: 1.02–1.12]) (S1 Table). Risk of LTFU was lower for those initiated during Covid-19 lockdowns relative to those initiated prior (S1 Table). This effect was consistent in the aggregated and sex-stratified analyses for after initiation (RR: 1.09 [95% CI: 1.04–1.14]) and within the first two years of treatment (RR: 2.01

**Table 2. Characteristics of current visits archetypes based on longitudinal patterns of ART visit attendance in a cohort of 191,162 patients initiating antiretroviral therapy in South Africa from Jan 2017-March 2022.**

| Visit archetype | | % of total visits | % Next visit IIT |
|---|---|---|---|
| Adherent | On time | 75.31% (N = 1,309,143) | 6.30% |
| Late | Late Once | 10.63% (N = 184,827) | 8.74% |
| | Late Twice | 2.89% (N = 50,280) | 11.85% |
| Interrupter | First time interrupter | 3.48% (N = 60,553) | 13.48% |
| | First time interrupter late previously | 0.81% (N = 14,154) | 17.54% |
| | Repeat interrupter | 0.96% (N = 16,639) | 15.22% |
| | Repeat interrupter late previously | 0.41% (N = 7,178) | 21.13% |
| | ITT twice | 0.82% (N = 14,276) | 23.48% |
| Returning defaulter | Visit after ITT on time | 3.53% (N = 61,291) | 11.15% |
| | Visit after ITT late | 1.15% (N = 20,022) | 16.91% |

[95% CI: 1.92–2.11]). Similarly, the risk for LTFU decreased over time on treatment (S1 Table) for both men and women (S1 Table).

## Longitudinal risk factors for IIT

**Distribution of visit archetypes.** During the first two years of treatment, 49% (N = 95,581) of patients who attended at least one additional visit after initiation experienced at least one IIT. Of the 1,778,074 visits observed, 75% were attended on time, 14% were attended late, 7% were defined as IIT and 4% of visits occurred after an interruption in treatment. Based on our operational definition of visits attendance (Fig 1), visits were classified by the mutually exclusive and collectively exhaustive visit archetypes defined in Table 2.

While the overall rate of return to treatment was lower than interruption in treatment (Fig 2), 85% of patients returned for a subsequent ART visit after an interruption in treatment. After an initial peak due to those who drop out of treatment after initiation, rates of IIT consistently increase with time on treatment with marked declines at the one year and two year time points (Fig 2).

As with LTFU, age at ART initiation and baseline CD4 count were not shown to be predictive of risk of IIT (Fig 3, S2 Table). Previous visit attendance was shown to be associated with the risk of experiencing an IIT with both late previous visit attendance and having had a historical IIT having increased hazards of the next visit being an IIT (Fig 2, S2 Table).

## Relating linear risk factors to machine learning model predictions

When using the original set of 13 predictors, model performance decreased relative to previous iterations [17] when trained and tested with health records collected during the and after Covid-19 lockdown measures (Table 3). While sensitivity remained similar, with both models able to predict approximately 62% of next missed visits, PPV decreased by 2% translating to 17.5% next visits labelled as missed to be truly missed (Table 3). Model performance was not improved with the addition of a singular categorical predictor describing previous visits archetypes (Table 3). In comparison to the full model containing information on historical visits attendance, baseline demographics and clinical features, a model using only previous visit archetypes was able to correctly predict almost half of next missed visits, with a small decrease in precision (PPV of 16.5%).

We ranked previous archetypes based on the hazard ratios calculated in S2 Table, and evaluated how these results related to the risk of IIT predicted by the CatBoost model (Fig 4). We

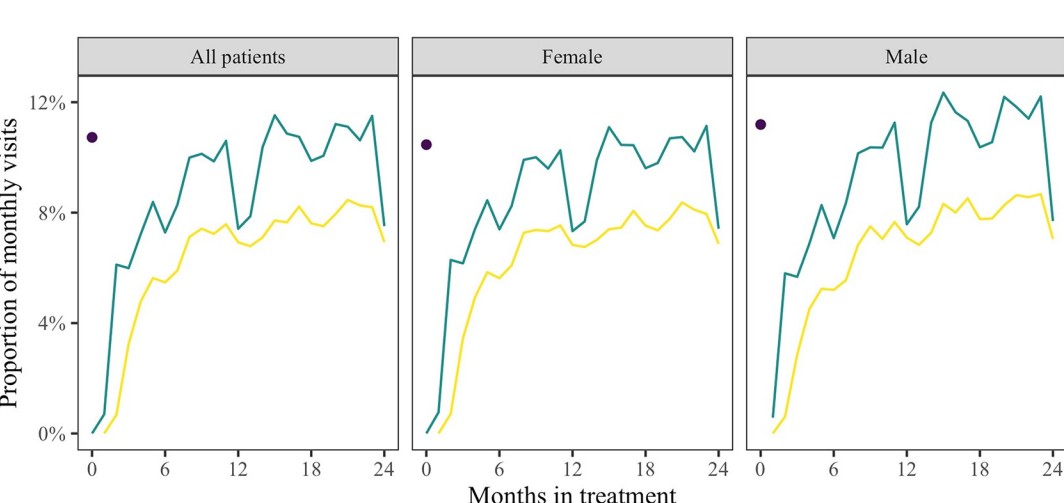

**Fig 2. Longitudinal ART clinic visit attendance in the first two years of antiretroviral therapy in a cohort of 191,162 patients initiating antiretroviral therapy in South Africa from Jan 2017-March 2022.** Purple dots represent the proportion of patients that do not return to treatment after treatment initiation. Green lines represent the monthly proportion of visits that are attended more than 28 days after a scheduled appointment (ITT) and yellow lines represent the proportion of monthly visits attended by patients within 28 days of a scheduled visit date after previously experiencing an ITT (Return after ITT).

| | **All patients** | | | **Females** | | | **Males** | | |
|---|---|---|---|---|---|---|---|---|---|
| **Months in treatment** | **0-6** | **6 -12** | **12-14** | **0-6** | **6 -12** | **12-14** | **0-6** | **6 -12** | **12-14** |
| **% Initiation dropout** | 2.14% (N=18,568) | | | 2.09% (N=11,595) | | | 2.23% (N=6973) | | |
| **Visits attended on time** | 91.59% (N=795,465) | 82.98% (N=503,009) | 84.62% (N=384,846) | 91.41% (N=507,960) | 83.4% (N=329,738) | 84.69% (N=249,710) | 91.9% (N=287,505) | 82.19% (N=173,271) | 84.49% (N=135,136) |
| **% ITT** | 4.03% (N=34,971) | 9.65% (N=58,483) | 8.67% (N=39,415) | 4.15% (N=23,066) | 9.34% (N=36,935) | 8.54% (N=25,175) | 3.81% (N=11,905) | 10.22% (N=21,548) | 8.9% (N=14240) |
| **% Return to treatment after ITT** | 2.25% (N=19,537) | 7.37% (N=44,683) | 6.71% (N=30,534) | 2.35% (N=13,086) | 7.25% (N=28,681) | 6.77% (N=19,966) | 2.06% (N=64,51) | 7.59% (N=16,002) | 6.61% (N=10,568) |
| **Total number visits** | 868,541 | 606,175 | 454,795 | 555,707 | 395,354 | 294,851 | 312,834 | 210,821 | 159,944 |

found that relative to previous on time visits, late previous visit attendance or historical IIT was associated with a prediction of IIT in the machine learning model. Subsequent late visit attendance and/or IITs were strongly associated with elevated risk of IIT in both the machine learning and adjusted cox models. While associated with a relatively smaller increased risk of IIT compared to other visit archetypes in the survival analysis, a late visit where a previous visit was an IIT often preceded an IIT in the machine learning predictions. Conversely, a single late visit, shown to confer an elevated 20% hazard of the occurrence of IIT in the survival analysis was not a strong predictor of IIT in the machine learning model.

## Discussion

ART patient retention is a priority for achieving epidemic control in the South African HIV epidemic. To design effective intervention strategies, there is a need for more precise descriptions of the longitudinal changes in the risk of LTFU as well as the characteristics of those who disengage from treatment [26]. We have previously reported that a machine learning model informed by historical visit attendance, baseline demographics and clinical risk factors is able to predict up to two-thirds of next missed ART clinic visits [17]. Here, we demonstrate that

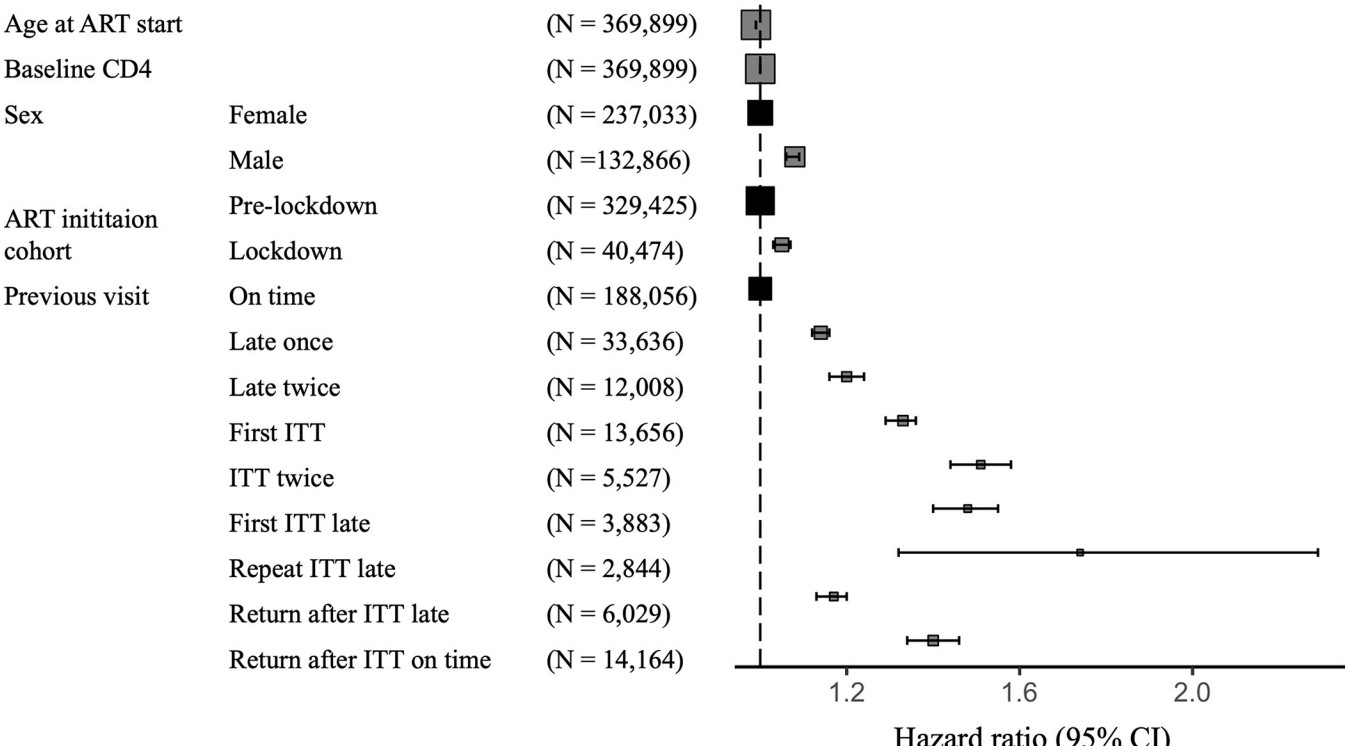

**Fig 3. Adjusted survival analysis of baseline and longitudinal risk factors for interruption in treatment in a cohort of 191,162 patients initiating antiretroviral therapy in South Africa from Jan 2017-March 2022.** Results from semi-parametric extension of the cox model are summarised as exponentiated hazards ratios (box) and 95% confidence intervals (whiskers). Colours denote reference (black) and comparator (grey) groups for categorical variables and size denotes number of observations in each variable group.

historical visit attendance alone is able to predict up to half of next missed visits and is predictive of IIT using both machine learning and traditional statistical methods.

Depending on the program context, the operational definition of LTFU is defined as anywhere between 28–90 days out of treatment [20, 22]. Given inconsistent definitions of LTFU, the inability to account for undocumented patient transfers between clinics and undocumented mortality, true rates of LTFU in South Africa are difficult to quantify [27, 28]. A cohort study involving intensive retrospective contact tracing of patients who discontinued ART at a Kwazulu-Natal clinic found that only 14% of patients marked as LTFU were truly unaccounted for [10]. Using a definition of 90 days out of treatment we observed that almost 40% of patients became LTFU within the first two years of treatment and 50% of treatment discontinuation occurred the first month of treatment. When assessing trends in IIT, we observed that 85% of patients who miss a visit by more than 28 days return for a subsequent visit, implying that cross-sectional estimates of LTFU are not a good indicator of current treatment coverage. We observed temporal landmarks in ART visit attendance, with rates of IIT being lowest at the 1-year and 2-year landmarks. This demonstrates that ART retention and treatment engagement are dynamic processes and current approaches that do not consider temporal trends are not appropriate for characterising gaps in care in over time [29]. Understanding this is critical in informing cross-sectional estimates of treatment coverage, given the large variation in the sensitivity and specificity of current methods to assess ART treatment adherence [30].

Collinearity and the non-linear nature of predictors in our previously validated machine learning model limit explainability of risk factors identified as predictive of IIT [17] and

**Table 3. Model performance in analysis of prediction of interruption in treatment in a cohort of 191,162 patients initiating antiretroviral therapy in South Africa from Jan 2017-March 2022.** We compared the performance of (A) the top 13 predictors from the validated CatBoost model, (B) the addition of previous visit archetypes to the validated CatBoost model and (C) a model using only previous visit archetypes as predictor.

| Model | | (A) CatBoost model: top 13 predictors | (B) CatBoost model: top 13 predictors + archetypes | (C) CatBoost model: Archetypes only |
|---|---|---|---|---|
| **Cohort** | | 1 Jan 2017 - | 1 Jan 2017 - | 1 Jan 2017 - |
| | | 31 March 2020 | 24 March 2022 | 24 March 2022 |
| **Patients** | | 136,082 | 191,162 | 191,162 |
| **Visits** | | 1,494,728 | 1,833,248 | 1,833,248 |
| | **Socio-demographic** | Age | Age | |
| | | Sex | Sex | |
| | **Clinical** | ART regimen duration | ART regimen duration | |
| | | Viral load count | Viral load count | |
| | | Last VL Value | Last VL Value | |
| | **Historical visit attendance** | Visit count | Visit count | |
| | | Months since last visit | Months since last visit | |
| | | Months since first visit | Months since first visit | |
| | | Next visit: day of month | Next visit: day of month | |
| | | Next visit: day of week | Next visit: day of week | |
| | | 3 days late ratio | 3 days late ratio | |
| | | 28 days late count | 28 days late count | |
| | | # Months missed Tx | # Months missed Tx | |
| | | | Previous visit archetype | Previous visit archetype |
| **N visits train (% ITT)** | | 1,833,248 (50%) | 4,638,562 (50%) | 4,638,562 (50%) |
| **N visits test (% ITT)** | | 456,472 (12%) | 456,472 (10%) | 456,472 (10%) |
| **Sensitivity** | | 61.9% (61.5–62.3%) | 62.4% (62.2–62.7%) | 48.1% (47.9–48.4%) |
| **Specificity** | | 66.5% (66.4–66.7%) | 66.6% (66.6–66.7%) | 72.4% (72.4–72.5%) |
| **PPV** | | 19.7% (19.5–19.9%) | 17.5% (17.4–17.6%) | 16.5% (16.4–16.6%) |
| **NPV** | | 93% (92.9–93%) | 94% (93.9–94%) | 92.5% (92.4–92.5%) |
| **F1 score** | | 0.299 (0.296–0.301) | 0.274 (0.272–0.275) | 0.246 (0.244–0.248) |
| **roc AUC** | | 0.692 (0.69–0.695) | 0.697 (0.695–0.698) | 0.625 (0.623–0.627) |

therefore the ability to understand drivers of risk at the individual level and intervene accordingly before a treatment interruption occurs. While the inclusion of demographic and clinical features improves model performance, we have demonstrated with both machine learning and traditional statistical methods, that historical visit attendance alone is a strong predictor of IIT. Furthermore, lateness and repeat patterns of lateness can predict IIT, irrespective of current age, age at ART initiation, sex and baseline CD4. The usage of machine learning models such as this are limited to clinical settings where infrastructure exist to extract EMR data and run computations in real-time. In settings where this is not possible, patient archetypes based on historical visits attendance may be used to triage patient retention interventions.

These findings are aligned with results observed in historical cohort studies reporting that the timeliness of clinic attendance is a good predictor of viral load suppression and the development of ART resistance [31, 32]. In the absence of observable risk factors, we believe lateness is an actionable behavioural flag for a patient that may become LTFU in future but is currently present at a healthcare access point. This finding may inform patient retention strategies by identifying patients who are good candidates for prioritised interventions—those who are demonstrating a willingness to be on treatment and experiencing some external barrier. This

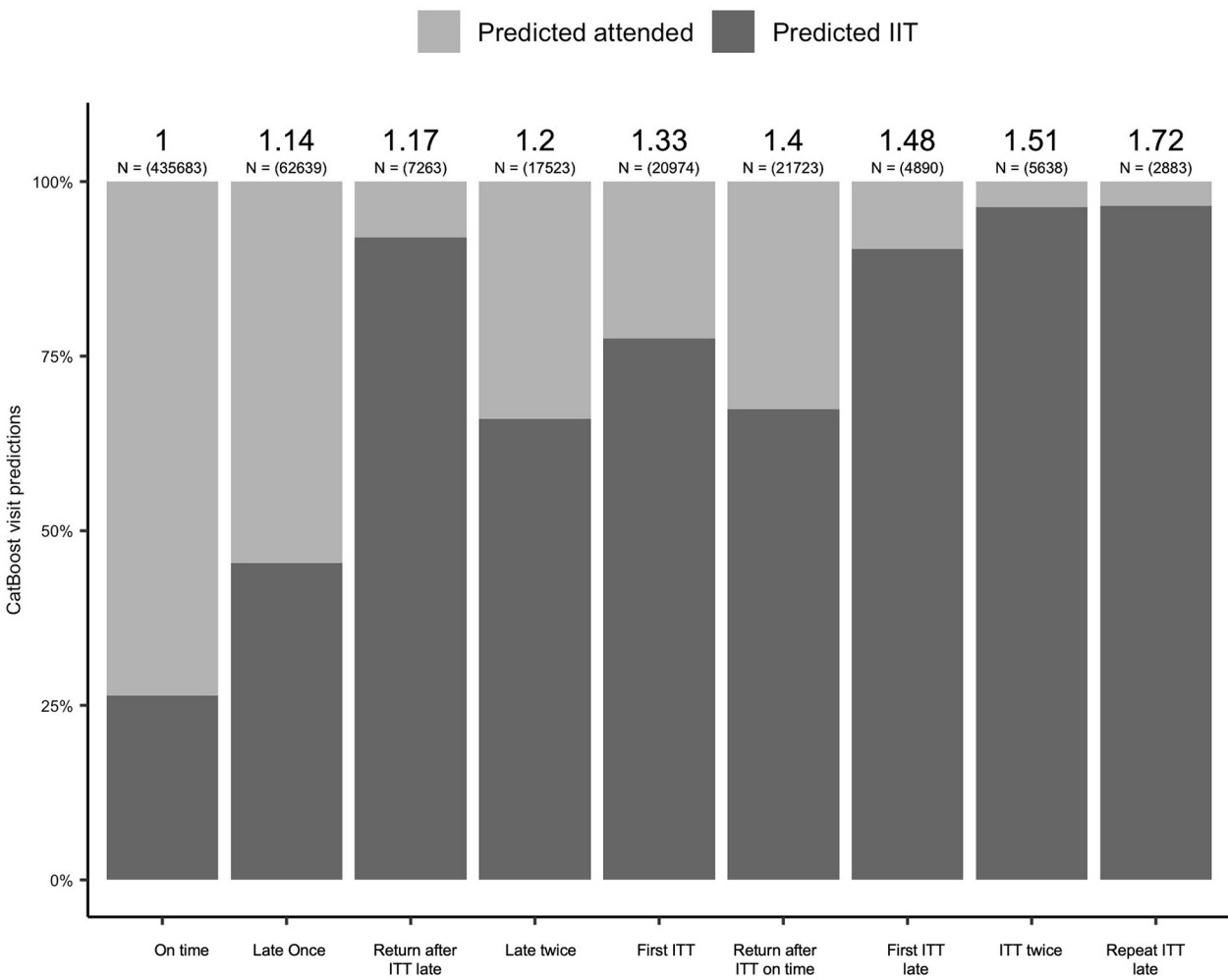

**Fig 4. CatBoost predicted probability of IIT for all visits summarised by previous visits type in a cohort of 191,162 patients initiating antiretroviral therapy in South Africa from Jan 2017-March 2022.** Hazard ratios for each visit archetype from adjusted survival model for all patients (Table 3) are labelled above.

creates the potential for targeted proactive intervention, as opposed to resource intensive retrospective tracing.

While there is little quantitative evidence detailing the effectiveness of individual retention interventions in South Africa [5, 6], modelling studies have demonstrated improving ART retention is cost saving even at low levels of effectiveness relative to alternatives for HIV spending [33]. A recent systematic review found that ART retention in South Africa was similar in standard care compared to 37 direct-service-delivery treatment interventions including those were facility-based individual models, out-of-facility-based individual models, client-led groups, and healthcare worker-led groups [5]. The direct effectiveness of retention interventions are difficult to quantify given that they are often implemented as part of multifaceted HIV service provision and retrospectively evaluated in an observational study framework [6]. While our work provides some visibility into ART patients who are at a high risk of LTFU, more work needs to be done in evaluating how the risk cohorts defined here can be effectively matched with appropriate retention or directed service delivery treatment modalities.

In previous applications of the IIT model, we censored data at the end of March 2020 as we were unable to account for interruptions in ART service delivery from the onset of the Covid-19 pandemic [17]. Here, we extended the application of the model to March 2022 and observed a moderate decrease in model performance despite doubling the size of the training dataset. Our regression analysis demonstrated that those initiated on ART during Covid lockdowns were at a reduced risk of longitudinal IIT and eventual LTFU relative to those initiated before. While this may reflect an improvement in treatment adherence, it is likely an artefact of the adoption of dispensing longer durations of ART treatment to account for limited facility access during that period [23, 34]. The sensitivity of the model to correctly predict IIT is a function of prevalence of the IIT as well as the occurrence of consistent patterns preceding IIT. Due to this, model development towards either sensitivity or precision is context dependent as has been discussed in our previous work [17]. Model performance may have been impacted by both a decrease in the overall observed rate of IIT and heterogeneity in visit attendance patterns after March 2020. Adding a single categorical predictor describing historical visit attendance did not improve model performance relative to the original set of predictors used, indicating that information on historical visit attendance is already distributed amongst the original set of predictors [17].

Due to the absence of unique patient identifiers, we were unable to account for patient mobility or validate outcome reporting. As a result, it is likely that a subset of patients classified as experiencing IIT or becoming LTFU were attending treatment at other facilities. Comparison of facility level outcomes to a South African national laboratory cohort demonstrates that HIV patient retention is underestimated at the facility level where undocumented patient transfers reflect as discontinuations in treatment [3]. Over six years of treatment, retention in care at the national level accounting for patient mobility was 63% relative to 29% at a facility level [3]. We plan on focusing future work on aligning viral load testing records with longitudinal patient attendance records as an improved method of ascertaining and predicting individual level treatment status.

A study of HIV patients in the United States demonstrated that data extracted from clinical records, patient mental health evaluations and insurance claims can be leveraged by machine learning methods to produce high precision predictions of patient behaviour across the HIV care cascade [35]. Without socio-behavioural information linked to routine HIV management systems, we use lateness as a signal for the occurrence events that increase the risk of IIT and LTFU. Using only longitudinal visits attendance and baseline clinical outcomes, we are able to predict two thirds of next missed visits. The incorporation of socio-behavioural data could improve the ability of this approach to inform retention interventions to prevent those at risk of disengaging from HIV care.

In this study, we describe baseline and time varying predictors of ART treatment in South African PLHIV. Longitudinal trajectories of ART visit attendance demonstrate that patients transition in and out of treatment indicating that patient retention in South Africa is likely underestimated. Historical visits attendance is predictive of future interruptions in treatment and can be used to identify those at risk of disengaging from HIV care in the absence of other behavioural or observable risk factors.

## Supporting information

**S1 Table. Log binomial regression results risk factors for LTFU in a cohort of 191,162 patients initiating antiretroviral therapy in South Africa from Jan 2017-March 2022.** (DOCX)

**S2 Table. Survival analysis of baseline and longitudinal risk factors for IIT in a cohort of 191,162 patients initiating antiretroviral therapy in South Africa from Jan 2017-March 2022.**
(DOCX)

## Author Contributions

**Conceptualization:** Rachel T. Esra.

**Data curation:** Michael Eisenstein.

**Formal analysis:** Rachel T. Esra, Jacques Carstens.

**Investigation:** Rachel T. Esra.

**Methodology:** Rachel T. Esra, Jacques Carstens, Lucien De Voux.

**Project administration:** Rachel T. Esra.

**Resources:** Rachel T. Esra.

**Software:** Rachel T. Esra, Jacques Carstens.

**Supervision:** Mhari Maskew.

**Validation:** Rachel T. Esra.

**Visualization:** Rachel T. Esra.

**Writing – original draft:** Rachel T. Esra.

**Writing – review & editing:** Jacques Carstens, Janne Estill, Ricky Stoch, Sue Le Roux, Tonderai Mabuto, Olivia Keiser, Mhari Maskew, Matthew P. Fox, Lucien De Voux, Kieran Sharpey-Schafer.

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
