## [Decision Letter · Decision Letter 0]

6 Mar 2023

PGPH-D-22-01941

Historical visit attendance as predictor of treatment interruption in South African HIV patients: extension of a validated machine learning model

Dear Dr. Esra,

Thank you for submitting your manuscript to PLOS Global Public Health. After careful consideration, we feel that it has merit but does not fully meet PLOS Global Public Health’s publication criteria as it currently stands. Therefore, we invite you to submit a revised version of the manuscript that addresses the points raised during the review process.

We look forward to receiving your revised manuscript.

Kind regards,

Hannah Hogan Leslie, PhD

Academic Editor

Journal Requirements:

1. Please ensure that Funding Information and Financial Disclosure Statement are matched.

2. In the Funding Information you indicated that no funding was received. Please revise the Funding Information field to reflect funding received.

4. In the online submission form, you indicated that "Access to primary data is subject to restrictions owing to privacy and ethics policies set by the South African Government. Code for the analysis may be accessed from the authors upon reasonable request". All PLOS journals now require all data underlying the findings described in their manuscript to be freely available to other researchers, either 1. In a public repository, 2. Within the manuscript itself, or 3. Uploaded as supplementary information.

Additional Editor Comments (if provided):

Reviewers' comments:

Reviewer's Responses to Questions

**Comments to the Author**

1. Does this manuscript meet PLOS Global Public Health’s publication criteria? Is the manuscript technically sound, and do the data support the conclusions? The manuscript must describe methodologically and ethically rigorous research with conclusions that are appropriately drawn based on the data presented.

Reviewer #1: Yes

Reviewer #2: Yes

2. Has the statistical analysis been performed appropriately and rigorously?

Reviewer #1: I don't know

Reviewer #2: Yes

3. Have the authors made all data underlying the findings in their manuscript fully available (please refer to the Data Availability Statement at the start of the manuscript PDF file)?

Reviewer #1: Yes

Reviewer #2: No

4. Is the manuscript presented in an intelligible fashion and written in standard English?

Reviewer #1: Yes

Reviewer #2: Yes

5. Review Comments to the Author

Reviewer #1: Overall comments:

I think this is an interesting manuscript to better understand the relationship between sociodemographic, clinical, and care history variables in patterns and prediction of retention. The authors use sophisticated methods (e.g., machine learning) and present interesting analyses to help unpack which variables are most important for understanding future risk of LTFU. I think this is an incredibly important question as to date most attention on risk prediction has focused primarily on sociodemographic and clinical variables rather than observed behavior like visit history. My main suggestions for this article are to better clarify the questions they are trying to answer with each analysis and why it is important. Currently, these are interesting analyses but the overall message, learning points are a bit obscured. I also have additional suggestions for potential analyses that may tighten the message and also to help with clarity.

Specific comments:

Intro

• I think the intro is generally well-written and touches on the important points, except I found the second paragraph and bit winding…it is more about how we target retention interventions rather than if they are effective or not.

• I think points that could be highlighted more clearly about how we target. A lot of attention is on just using clinical and sociodemographic characteristics that are not that predictive, but are easily identified. Statistical approaches that include more variables may be a bit better, but are harder to implement (and machine learning may be even a little bit better). But it seems to me that the most important aspect is what kind of data/variables we focus on and put into our models. This is discussed well in the discussion, but I think the main point is that we need to incorporate patients’ observed behaviors (i.e., their care history) into how we do risk assessments. Whether simple variables of care history (prior LTFU, prior lateness) suffice or more advanced machine learning algorithms are needed is I think an important point (and interesting question). I think some highlighting of both the types of data we use and also how sophisticated the algorithm should be discussed.

• For the last paragraph, I wasn’t totally clear on the purpose of the paper. Is it to identify variables that may improve a machine learning algorithm. Is it to identify which variables are most important (sociodemographics, clinical, care history).

• I would also distinguish between sociodemographic, clinical, care history predictors. I think many people just think of sociodemographic and clinical being important, but what this paper and the authors’ prior work really demonstrates is that care history is likely most important.

Methods:

• I had a bit of hard time keeping track of the definitions and how the analysis was being down. From my understanding, all analyses were done at the level of a scheduled appointment (whether they attended the visit or not). And then each visit (whether attended or not) were classified as on time, late, or IIT depending on when they eventually showed up (and then further categorized based on prior histories). And clarify at what level the analyses are being done.

•

• Figure 1 may need a more descriptive legend. To me I think the visit in gray is the one being categorized and the red, orange, green represent what has happened in the past (or perhaps it is the rightmost colored dot). A figure like this is great, but I think needs to be a lot clearer.

• How can some one return to treatment on time or late? If they have an appointment and are 28 days late they have an interruption, but they may come back at any time after 28 days late and reengage in care. Either way they are more than 28 days late for their last appointment. Looking more in depth at Table 1, the description also makes it sound like this is the visit after someone returns (not their return visit)…but then not sure how their return visit get classified (IIT?).

• For Cox PH model, would clarify what time zero is and also event. Is this time to first IIT or LTFU. Or this a recurrent event analysis looking forward from each made visit.

• How do the original 13 variables different from visit archetypes. My understanding is that previous history was already included in the original machine learning model (and those models may more flexibly characterize complex trajectories over time)…I do understand more after looking at Table 2 but would be nice to understand in methods so questsion becomes clear.

• In general, would try to be specific about the question each analysis is trying to answer. My sense is that you are trying to see whether visit archetypes are good enough and then don’t even require machine learning (or something to that end)…but I am not sure. Similar to machine learning vs. Cox PH

Results

• Should you also describe characteristics of visits as I think most analyses are being on that level?

• What does “returned to treatment after initiation” mean. I think it represent someone who came back a second time (i.e., not LTFU the day of initiation) but also sounds a bit like reengagement. May want to rephrase.

• I don’t fully understand Figure 2 and how the dynamics of LTFU and then return to care play out with the blue and yellow lines (related to prior comment).

• Table 2 seems to have some of the most interesting findings from the study. I think would be really interesting to step and consider the important to ask here, and I think that has to do with the types of variables included. I see 3 categories: sociodemographic (age, sex); clinical (VLs), but then more are related to care history (duration in care, appointments, lateness ratio, visit archetypes). Could examine what prediction looks like with or without care history (e.g., how good does sociodemographics do). Could examine using visit archetypes vs. the other nuanced care history variables. Ultimately, I think it comes down to what is most useful. A clinician can easily identify sex, age, and a visit archetype, but can’t really calculate lateness ratios (etc.). Could try and understand what is a minimum feasible to get good prediction.

Discussion

• I think the discussion is good and focuses on the important points, particularly the important of care history. I think a sharper focus on the questions at hand could be helpful (but I think will come from addressing my comments in other sections).

• Additional citations to consider for intro or discussion are PMID 33948789, 33105396, 32173743, 31661487. For disclosure, I am a co-author on these studies so feel free to include or not include based on your judgement. I think they may complement what this study does by discussing how we think about risk of LTFU, its time-varying nature, and what types are data needed to get better understandings, which I think are all very important questions that this paper helps unpack.

Reviewer #2: Background

Additionally, it would be helpful to clarify the specific research questions or hypotheses being addressed in this study. While the overall aim of the study is clear, it may be useful to explicitly state the research questions or hypotheses that were tested using the machine learning and traditional statistical methods.

Methods

The methods section could benefit from a clearer explanation of what "previous visit archetypes" are and how they were used in the analysis.

Overall

Overall, the manuscript appears to be well-written and provides valuable insights into patient retention in HIV treatment in South Africa. The use of machine learning to predict patient outcomes based on historical data is particularly innovative and could have important implications for improving patient retention strategies. However, the manuscript could benefit from additional detail on the specific interventions that could be implemented based on the findings. Additionally, further discussion of potential limitations of the study, such as the lack of unique patient identifiers and the potential for misclassification of patient outcomes, could help contextualize the results.

6. PLOS authors have the option to publish the peer review history of their article (what does this mean?). If published, this will include your full peer review and any attached files.

**Do you want your identity to be public for this peer review?** For information about this choice, including consent withdrawal, please see our Privacy Policy.

Reviewer #1: **Yes: **Aaloke Mody

Reviewer #2: No

---

## [Decision Letter · Decision Letter 1]

24 Apr 2023

PGPH-D-22-01941R1

Historical visit attendance as predictor of treatment interruption in South African HIV patients: extension of a validated machine learning model

Dear Dr. Esra,

Thank you for submitting your manuscript to PLOS Global Public Health. After careful consideration, we feel that it has merit but does not fully meet PLOS Global Public Health’s publication criteria as it currently stands. Therefore, we invite you to submit a revised version of the manuscript that addresses the points raised during the review process. In particular, reviewer concerns around the clarity of the research question, the transparency in decision making, and the conceptual basis for including variables have not been fully addressed within the manuscript. A more complete response is necessary to fairly evaluate the manuscript and progress it further. 

We look forward to receiving your revised manuscript.

Kind regards,

Hannah Hogan Leslie, PhD

Academic Editor

Journal Requirements:

Additional Editor Comments (if provided):

Reviewers' comments:

Reviewer's Responses to Questions

**Comments to the Author**

1. If the authors have adequately addressed your comments raised in a previous round of review and you feel that this manuscript is now acceptable for publication, you may indicate that here to bypass the “Comments to the Author” section, enter your conflict of interest statement in the “Confidential to Editor” section, and submit your "Accept" recommendation.

Reviewer #1: (No Response)

Reviewer #2: All comments have been addressed

2. Does this manuscript meet PLOS Global Public Health’s publication criteria? Is the manuscript technically sound, and do the data support the conclusions? The manuscript must describe methodologically and ethically rigorous research with conclusions that are appropriately drawn based on the data presented.

Reviewer #1: Yes

Reviewer #2: Yes

3. Has the statistical analysis been performed appropriately and rigorously?

Reviewer #1: Yes

Reviewer #2: Yes

4. Have the authors made all data underlying the findings in their manuscript fully available (please refer to the Data Availability Statement at the start of the manuscript PDF file)?

Reviewer #1: Yes

Reviewer #2: Yes

5. Is the manuscript presented in an intelligible fashion and written in standard English?

Reviewer #1: Yes

Reviewer #2: Yes

6. Review Comments to the Author

Reviewer #1: Overall comments:

Thank you for the authors’ for addressing my previous comments. I still had a few points for clarification. I still there is additional clarity around the question at hand and why its important.

• I would remove all references to causality. The issue with causal inference isn’t a statistical one due to non-linearity and collinearity. There are more complex issues at hand with time-varying mediation-confounding, etc. and unmeasured confounding that can’t be handled statistically.

• I agree with the author’s that discussion of the effectiveness of LTFU interventions is relevant, but as a reader, the narrative gets lost. Its not until the end of the intro or beginning of methods that I start to understand what the manuscript is about. Would just consider reframing/organizing so the main narrative stays clear.

• I appreciate the authors’ perspective that discussion of what variables are included are not within the scope of this manuscript, though I am not sure I agree. Again, this may be partially because I don’t follow the intro narrative for the specific questions and their implications. Machine learning also seems potentially useful when data is complex and would require a lot of decision-making beforehand. Considering these issues also directly guides what types of statistical issues and questions are most important to address.

• I still think for Table 3 would be important to consider the conceptually different categories for data types. Looking at the previously published paper, variables on visit history were the most important, sociodemographics less so. Is the questions here, will adding archetypes—when you already included other visit history in the machine learning model—improve prediction? And the rationale for adding archetypes is because they were associated with outcomes using more routine statistical methods (although those models didn’t include visit history). Again I may not be following correctly, but I would not anticipate that would add much. There are many practical considerations to consider to how one would want to add additional variables like visit archetypes to an existing model. I don’t particularly follow the rationale for the approach taken here. Some conceptual distinction between the variables used in the prior models vs these archetypes should be made (outside of the history of how the authors’ have chosen to use them).

Reviewer #2: Thanks for the thorough review and making the manuscrit clear.

7. PLOS authors have the option to publish the peer review history of their article (what does this mean?). If published, this will include your full peer review and any attached files.

**Do you want your identity to be public for this peer review?** For information about this choice, including consent withdrawal, please see our Privacy Policy.

Reviewer #1: **Yes: **Aaloke Mody

Reviewer #2: No

---

## [Decision Letter · Decision Letter 2]

22 May 2023

PGPH-D-22-01941R2

Historical visit attendance as predictor of treatment interruption in South African HIV patients: extension of a validated machine learning model

Dear Dr. Esra,

Thank you for submitting your manuscript to PLOS Global Public Health. After careful consideration, we feel that it has merit but does not fully meet PLOS Global Public Health’s publication criteria as it currently stands. Therefore, we invite you to submit a revised version of the manuscript that addresses the points raised during the review process. Thank you for addressing the reviewer's comments on the previous submission; the revisions have substantially clarified the paper. I would ask that you expand on the findings in Table 3 and link these to the summary implications in the discussion and abstract to be sure the clarified research aims are fully executed in the paper.

We look forward to receiving your revised manuscript.

Kind regards,

Hannah Hogan Leslie, PhD

Academic Editor

Journal Requirements:

Additional Editor Comments (if provided):

There is limited engagement in the text with the contents of Table 3, which is the key evidence in response to the question on whether categorical visit attendance is on its own a valid predictor. The discussion summarizes across the models ("Historical visit attendance is highly informative in a machine learning model predicting up to two thirds of next missed ART visits" - presumably a reference to the sensitivity >60% of the 2 models with clinical and demographic variables also included) and the abstract highlights the performance of the model without categorical adherence. Please provide greater interpretation in the text of the results shown in Table 3 and elaborate in the discussion what the implications are for both variable type and statistical approach recommended for use moving forward.

Reviewers' comments:

Reviewer's Responses to Questions

**Comments to the Author**

1. If the authors have adequately addressed your comments raised in a previous round of review and you feel that this manuscript is now acceptable for publication, you may indicate that here to bypass the “Comments to the Author” section, enter your conflict of interest statement in the “Confidential to Editor” section, and submit your "Accept" recommendation.

Reviewer #1: (No Response)

2. Does this manuscript meet PLOS Global Public Health’s publication criteria? Is the manuscript technically sound, and do the data support the conclusions? The manuscript must describe methodologically and ethically rigorous research with conclusions that are appropriately drawn based on the data presented.

Reviewer #1: Yes

3. Has the statistical analysis been performed appropriately and rigorously?

Reviewer #1: Yes

4. Have the authors made all data underlying the findings in their manuscript fully available (please refer to the Data Availability Statement at the start of the manuscript PDF file)?

Reviewer #1: Yes

5. Is the manuscript presented in an intelligible fashion and written in standard English?

Reviewer #1: Yes

6. Review Comments to the Author

Reviewer #1: Thank you for the authors’ for addressing my previous comments. I think the manuscript is improved and easier to follow with these changes. I do still find some of the explanation dense, particularly related to Table 3, and I feel a general reader who is not familiar with differences between examining causality vs. prediction would struggle. Ultimately, what I believe the authors are saying and an underlying point is that machine learning algorithms are complex and model complex relationships, thus they are not readily useable in clinical settings (unless implemented in an EHR that can do the computations in the real-time in the background). Essentially, they are a black box. Visit archetypes, however, are readily identifiable by clinicians/practitioners—either with EHRs or not—and without any need for understanding complex relationships between variables. Thus, it is useful and important to understand the performance of these archetypes. I think bringing out these practical implications would be helpful in the results and discussion.

7. PLOS authors have the option to publish the peer review history of their article (what does this mean?). If published, this will include your full peer review and any attached files.

**Do you want your identity to be public for this peer review?** For information about this choice, including consent withdrawal, please see our Privacy Policy.

Reviewer #1: No

---

## [Editor Report · Decision Letter 3]

6 Jun 2023

Historical visit attendance as predictor of treatment interruption in South African HIV patients: extension of a validated machine learning model

PGPH-D-22-01941R3

Dear Ms Esra,

We are pleased to inform you that your manuscript 'Historical visit attendance as predictor of treatment interruption in South African HIV patients: extension of a validated machine learning model' has been provisionally accepted for publication in PLOS Global Public Health.

Best regards,

Hannah Hogan Leslie, PhD

Academic Editor